# LEARNING INDEPENDENT CAUSAL MECHANISMS

## ABSTRACT

Independent causal mechanisms are a central concept in the study of causality with implications for machine learning tasks. In this work we develop an algorithm to re-cover a set of (inverse) independent mechanisms relating a distribution transformed by the mechanisms to a reference distribution. The approach is fully unsupervised and based on a set of experts that compete for data to specialize and extract the mechanisms. We test and analyze the proposed method on a series of experiments based on image transformations. Each expert successfully maps a subset of the transformed data to the original domain, and the learned mechanisms generalize to other domains. We discuss implications for domain transfer and links to recent trends in generative modeling.

## 1 INTRODUCTION

Humans are able to recognize objects such as handwritten digits based on distorted inputs. When presented with digits which are translated, corrupted, or inverted, we can usually correctly label them without the need of re-learning them from scratch. The same applies for new objects, essentially after having seen them once. This may be due to the fact that human intelligence utilizes *mechanisms* (such as translation) that are generic and *generalize* across object classes. These mechanisms are *modular, re-usable and broadly applicable*, and the problem of learning them from data is fundamental for the study of transfer.

In the field of causality, the concept of independent mechanisms plays a central role both on the conceptual level and, more recently, in applications to inference. The *independent mechanism (IM)* assumption states that the causal generative process of a system's variables is composed of autonomous modules that do not inform or influence each other (Schölkopf et al., 2012; Peters et al., 2017).

If a joint density is Markovian with respect to a directed graph $\mathcal{G}$, we can write it as

$$p(\mathbf{x}) = p(x_1, \ldots, x_d) = \prod_{j=1}^{d} p(x_j | \mathrm{pa}_{\mathcal{G}}^j), \tag{1}$$

where $\mathrm{pa}_{\mathcal{G}}^j$ denotes the parents of variable $x_j$ in the graph.

For a given joint density, there are usually many decompositions of the form (1), with respect to different graphs. If $\mathcal{G}$ is a *causal* graph, i.e., if its edges denote direct causation (Pearl, 2000), then the conditional $p(x_j | \mathrm{pa}_{\mathcal{G}}{}^j)$ can be thought of as physical *mechanism* generating $x_j$ from its parents, and we refer to it as a *causal conditional*. In this case, we consider (1) a *generative* model where the term "generative" truly refers to a physical generative process. As an aside, we note that in the alternative view of causal models as structural equation models, each of the causal conditionals corresponds to a functional mapping and a noise variable (Pearl, 2000).

By the IM assumption, the causal conditionals are autonomous modules that do not influence or inform each other. This has multiple consequences. First, knowledge of one mechanism does not contain information about another one (Appendix D). Second, if one mechanism changes (e.g., due to distribution shift), there is no reason that other mechanisms should also change, i.e., they tend to remain *invariant*. As a special case, it is (in principle) possible to locally *intervene* on one mechanism (for instance, by setting it to a constant) without affecting any of the other modules. In all these cases, most of (1) will remain unchanged. However, since the overall density will change, in the generic case the (non-causal) conditionals would change.

The IM assumption can be exploited when performing causal structure inference (Peters et al., 2017). However, it also has implications for machine learning more broadly. A model which is expressed in terms of causal conditionals (rather than conditionals with respect to some other factorization) is likely to have components that better transfer or generalize to other settings (Schölkopf et al., 2012), and its modules are better suited for building complex models from simpler ones. Independent modules as sub-components can be trained independently, from multiple domains, are more likely to be re-usable. They can also be easier to interpret since they correspond to physical mechanisms. Animate intelligence cannot afford to learn new models from scratch for every new task. Rather, it is likely to rely on robust local components that can flexibly be re-used and re-purposed. It also requires local mechanisms for adapting and training modules rather than re-training the whole brain every time a new task is learned.

In the present paper, we focus on a class of such modules, and on algorithms to learn them from data. We describe an architecture using competing experts specializing on different transformations. The resulting model permits a form of lifelong learning, with the possibility of easily adding, removing, retraining, and exporting its components independently.

In line with the intuition given above, we illustrate our approach on MNIST digits which have undergone different transformations such as contrast inversion, noise addition and translation. Information about the nature and number of such transformations need not be known at the beginning of training. Our goal is to identify the independent mechanisms linking a reference distribution to a distribution of modified digits, and learn to invert them without supervision. The inverse mechanisms can be used to transform modified digits and classify them using a standard MNIST classifier, thus exhibiting a form of robustness that animate intelligence excels at.

## 2 Related work

Our work mainly draws from mixtures of experts, domain adaptation, and causality.

Early works on mixture of experts date back to the early nineties (Jacobs et al. (1991), Jordan & Jacobs (1994)), and since then the topic has been subject of extensive research. Recent work include Shazeer et al. (2017), where the authors train a mixture of 1000 experts using a gating mechanism that selects only a very small number of experts for each example, and propose several technical solutions to deal with model and data parallelism. Aljundi et al. (2016) train a network of experts on multiple tasks, with a focus on lifelong learning; autoencoders are trained for each task and used as gating mechanisms.

Another research direction that is relevant to our work is unsupervised domain adaptation (Bousmalis et al., 2016). These methods often use some supervision from labeled data and/or match the two distributions in a learned feature space (Tzeng et al., 2017, e.g.).

The novelty of our work lies in the following aspects: (1) we automatically identify and invert a set of independent (inverse) causal mechanisms; (2) we do so using only data from an original distribution and from the mixture of transformed data, without labels; (3) the architecture is modular, can be easily expanded, and its trained modules can be reused; and (4) the method relies on competition of experts.

Ideas from the field of causal inference inspire the present work. Understanding the data generating mechanisms plays a key role in causal inference, and goes beyond the statistical assumptions usually exploited in machine learning. Causality provides a framework for understanding how a system responds to *interventions*, and causal graphical models as well as structural equation models (SEM) are common ways of describing causal systems (Pearl, 2000; Peters et al., 2017). The IM assumption discussed in the introduction can be used for identification of causal models (Daniušis et al., 2010; Zhang et al., 2015), but causality has also proven a useful tool for discussing and understanding machine learning in the non-i.i.d. regime. Recent applications include semi-supervised learning (Schölkopf et al., 2012) and transfer learning (Rojas-Carulla et al., 2015), in which the authors focus only on linear regression models. We seek to extend applications of causal inference to more complex settings and aim to learn causal mechanisms and ultimately causal SEMs without supervision.

There are close relations between our setting and recent work on deep learning for disentangling factors of variation (Chen et al., 2016; Higgins et al., 2016) as well as non-linear ICA (Hyvarinen & Morioka, 2016). In our work, causal mechanisms play the role of factors of variation. The main difference is that we currently recover inverse mechanisms as independent modular parts, instead of indentifying a joint low dimensional representation of the data without explicit separate paths for each factor.

## 3 LEARNING CAUSAL MECHANISMS AS INDEPENDENT MODULES

The aim of this section is twofold. First, we describe the generative process of our data. We start with a distribution $P$ that we will call "canonical" and an a priori *unknown* number of independent mechanisms which act on (examples drawn from) $P$. At training time, a sample from the canonical distribution is available, as well as a dataset obtained by applying the mechanisms to (unseen) examples drawn from $P$. Second, we propose an algorithm which recovers and learns to invert the mechanisms in an *unsupervised fashion*.

### 3.1 FORMAL SETTING

Consider a canonical distribution $P$ on $\mathbb{R}^d$, e.g., the empirical distribution defined by MNIST digits on pixel space. We further consider $N$ measurable functions $M_1, \ldots, M_N : \mathbb{R}^d \to \mathbb{R}^d$, called *mechanisms*. We think of these as independent causal mechanisms in nature, and their number is a priori unknown. A more formal definition of independence between mechanisms is relegated to Appendix D. The mechanisms give rise to $N$ distributions $Q_1, \ldots, Q_N$ where $Q_j = M_j(P)$.[1] In the MNIST example, we consider translations or adding noise as mechanisms, i.e., the corresponding $Q$ distributions are translated and noisy MNIST digits.

At training time, we receive a dataset $\mathcal{D}_Q = (x_i)_{i=1}^n$ drawn i.i.d. from a mixture of $Q_1, \ldots, Q_N$, and an independent sample $\mathcal{D}_P$ from the canonical distribution $P$. Our goal is to identify the underlying mechanisms $M_1, \ldots, M_N$ and learn approximate inverse mappings which allow us to map the examples from $\mathcal{D}_Q$ back to their counterpart drawn from $P$.

If we were given distinct datasets $\mathcal{D}_{Q_j}$ each drawn from $Q_j$, we could individually learn each mechanism, resulting in independent approximations regardless of the properties of the training procedure. This is due to the fact that the datasets are drawn from independent mechanisms and the separate training procedure cannot generate a dependence between them. This property is independent of properties of training, and does not require that the procedure is successful, i.e., that the obtained mechanisms approximate the true $M_j$ in some metric. In our case, we do not have access to the distinct datasets. Instead we construct a larger set $\mathcal{D}_Q$ by first taking the union of the sets $D_{Q_j}$, and then applying a random permutation. This corresponds to a dataset where each element has been generated by one of the (independent) mechanisms, but we don't know by which one. Clearly, it should be harder to identify and learn independent mechanisms from such a dataset. This is the setting we address below, and the crucial idea will be that of *competition*.

### 3.2 COMPETITIVE LEARNING OF INDEPENDENT MECHANISMS

In this section, we introduce our training protocol to address the problem defined above.

The training machine is composed of $N'$ parametric functions $E_1, \ldots, E_{N'}$ with distinct trainable parameters $\theta_1, \ldots, \theta_{N'}$. We refer to these functions as the *experts*. Note that we do not require $N' = N$, since the real number of mechanisms is unknown a priori. The goal is to maximize an objective function $c : \mathbb{R}^d \to \mathbb{R}$ with the key property that $c$ takes high values on the support of the canonical distribution $P$, and low values outside. Note that it is possible for $c$ to be a parametric function, and for these parameters to be jointly optimized with the experts during training. Below, we specify the details of this rather general definition.

During training, the experts compete for the data points. Each example $x'$ from $\mathcal{D}_Q$ is fed to all experts independently and in parallel. Depending on the output of each expert $c_j = c(E_j(x'))$, we select the winning expert $E_{j^*}$, where $j^* = \arg\max_j(c_j)$. $E_{j^*}$ *wins* the example $x'$, and its

---

[1]Each distribution $Q_j$ is defined as the pushforward measure induced by $P$ via $M_j$.

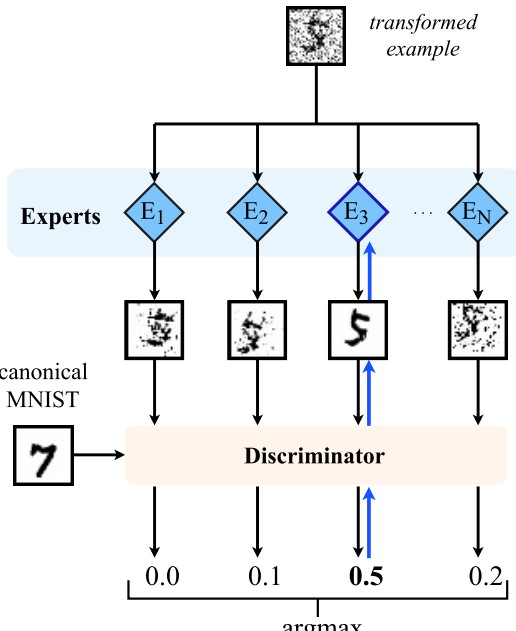

Figure 1: We show how a transformed example, here a noisy digit, is processed by a competition of experts. Only Expert 3 is specializing on denoising, it wins the example and gets trained on it, whereas the others perform translations and are not updated.

parameters $\theta_{j*}$ are updated as to maximize $c(E_{j*}(x'))$, while the other experts remain unchanged. The motivation behind competitively updating only the winning expert is to enforce specialization; the best performing expert becomes even better at mapping $x'$ back to the corresponding sample from the canonical distribution. Figure 1 depicts this procedure. Overall, our optimization problem reads

$$\theta_1^*, \ldots, \theta_{N'}^* = \underset{\theta_1,\ldots,\theta_{N'}}{\arg\max} \, \mathbb{E}_{x' \sim Q} \left( \max_{j \in \{1,\ldots,N'\}} c(E_{\theta_j}(x')) \right). \tag{2}$$

The training described above raises a number of questions, which we address next.

**1. Convergence criterion.** Since the problem is fully unsupervised, there is no straightforward way of measuring convergence, which raises the question of how to choose a stopping time for the competitive procedure. As an example, one may act according to one of the following: *a*) fix a maximum number of iterations or *b*) stop if each example is assigned to the same experts for a pre-defined number of iterations (i.e., each expert consistently wins the same data points).

**2. Selecting the appropriate number of experts.** Generally, the number of mechanisms $N$ which generated the dataset $\mathcal{D}_Q$ is not available a priori. Therefore, it is important to develop an adaptive procedure for setting up the number of experts $N'$. This is a common problem shared with most clustering techniques. Given the modular behavior of the procedure, experts may be added or removed during or after training, making the framework very flexible. Assuming however that the number of experts is fixed, we speculate that the following behaviors are likely.

If $N' > N$ (too many experts): a) some of the experts do not specialize and do not win any example in the dataset; or b) some tasks are divided between experts (for instance, each expert can specialize in a mode of the distribution of the same task). In a), the inactive experts can be removed, and in b) experts sharing the same task can be merged into a wider expert.[2]

If $N' < N$ (too few experts): a) some of the experts specialize in multiple tasks or b) some of the tasks are not learned by the experts, so that data points from such tasks lead to a poor score across all experts.

---

[2]However, note that in order to do this, it is necessary to first acknowledge that the two experts have learned part of the same task, which would require extra information or visual inspection.

While these questions are relevant, we do not develop them in detail and leave them for further research. Some experiments substantiating these claims can be found in Appendix A.2.

**3. Time and space complexity.** Each example has to be evaluated by all experts in order to assign it to the winning expert. While this results in a computational cost that depends linearly on the number of experts, these evaluations can be done in parallel and therefore the time complexity of a single iteration can be bounded by the complexity to compute the output of a single expert. Moreover, as each expert will in principle have a smaller architecture than a single large network, the committee of experts will typically be faster to execute.

**Concrete protocol for neural networks.** One possible model class for the experts are deep neural networks. Training using backpropagation is particularly well suited for the online nature of the training proposed: after an expert wins a data point $x'$, its parameters are updated by backpropagation, while other experts remain untouched. Moreover, recent advances in generative modeling give rise to natural choices for the loss function $c$. For instance, given a variational autoencoder (VAE) (Kingma & Welling, 2013) trained on the canonical distribution $P$, one may define $c(x')$ as the opposite of the VAE loss. The assumption is that the loss will only be low for examples drawn from $P$. Another possibility is to use adversarial training (Goodfellow et al., 2014), and use as an objective function the output of a discriminator network trained on the canonical sample $\mathcal{D}_P$ and against the outputs of the experts. In the next section we introduce a formal description of a training procedure based on adversarial training in Algorithm 1, and present experimental evidence of its good performance.

## 4 EXPERIMENTS

In this set of experiments we test the method presented in Section 3 on the MNIST dataset transformed with the set of mechanisms described in detail in the Appendix C, i.e. eight directions of translations by 4 pixels (up, down, left, right, and the four diagonals), contrast inversion, addition of noise, for a total of 10 transformations. We split the training partition of MNIST in half, and transform all and only the examples in the first half: this ensures that there is no matching ground truth for the experts to learn the mechanisms, and that learning is fully unsupervised. As a preprocessing step, the digits are zero-padded so that they have size $32 \times 32$ pixels, and the pixel intensities are scaled between 0 and 1. This is done even before any mechanism is applied. We use deep neural networks for both the experts and the selection mechanism, and use an adversarial training scheme.

Each expert $E_i$ can be seen as a generator from a GAN, that is conditioned on an input image instead of (or in addition to) a noise vector. A discriminator $D$ provides gradients for training the experts and acts also as a selection mechanism $c$: only the expert whose output obtains the higher score from $D$ *wins* the example, and is trained on it to maximize the output of $D$. We describe the exact algorithm used to train the networks in these experiments in Algorithm 1. The discriminator is trained to maximize the following cross-entropy loss:

$$\max_{\theta_D} \left( \mathbb{E}_{x \sim P} \log(D_{\theta_D}(x)) + \frac{1}{N'} \sum_{j=1}^{N'} \mathbb{E}_{x' \sim Q} \left( \log(1 - D_{\theta_D}(E_{\theta_j}(x'))) \right) \right) \tag{3}$$

For simplicity, we assume for the rest of this section that the number of experts $N'$ equals the number of true mechanisms $N$. Results where $N \neq N'$ are relegated to Appendix A.2.

**Neural nets details.** Each expert is a CNN with five convolutional layers, 32 filters per layer of size $3 \times 3$, ELU (Clevert et al. (2015)) as activation function, batch normalization (Ioffe & Szegedy (2015)), and zero padding. The discriminator is also a CNN, with average pooling every two convolutional layers, growing number of filters, and a fully connected layer with 1024 neurons as last hidden layer. Both networks are trained using Adam as optimizer (Kingma & Ba (2014)), with the default hyper-parameters.[3]

Unless specified otherwise, after a random weight initialization we first train the experts to approximate the identity mapping on our data, by pretraining them for up to 200 iterations on predicting

---

[3]For the exact experimental parameters and architectures see the Appendix B or the Pytorch implementation that will be released online.

---

**Algorithm 1** Learning independent mechanisms using competition of experts and adversarial training

---

**Precondition:** $X$: data sampled from $P$; $X'$: data sampled from $\mathcal{D}_Q$; $D$ discriminator; $N'$: number of experts; $T$: maximum number of iterations;
   **(p)** highlights that the steps in the instruction can be executed in parallel

1  $\{E_i \leftarrow \text{TrainNewAutoencoderOn}(X')\}_{j=1}^{N'}$            ▷ Init set of experts as approx identity **(p)**
2  **for** $t \leftarrow 1$ to $T$ **do**
3      $x, x' \leftarrow \text{Sample}(X), \text{Sample}(X')$                    ▷ Sample minibatches
4      $\{c_j \leftarrow D(E_j(x'))\}_{j=1}^{N'}$              ▷ Scores from $D$ for all outputs from the experts **(p)**
5      $\theta_D^{t+1} \leftarrow \text{Adam}(\theta_D^t, \nabla \log D(x) + \nabla(1/N' \sum_{j=1}^{N'} \log(1 - c_j)))$      ▷ Update $D$ **(p)**
6      $\{\theta_{E_j}^t \leftarrow \text{Adam}(\theta_{E_j}^t, \nabla \max_{j \in 1,\ldots,N'} \log(c_j))\}_{j=1}^{N'}$        ▷ Update experts **(p)**

---

identical input-output pairs randomly selected from the transformed dataset. This makes the experts start from similar grounds, and we found that this improved the speed and robustness of convergence. We will refer to this as *approximate identity initialization* for the rest of the paper.

A minibatch of 32 transformed MNIST digits, each transformed by a randomly chosen mechanism, is fed to all experts $E_i$. The outputs are fed to the discriminator $D$, which computes a score for each of them. For each example the cross entropy loss in Equation (3) and the resulting gradients are computed only for the output of the highest scoring expert, and they are used to update both the discriminator (when 0 is the target in the cross entropy) and the winning expert (when using 1 as the target). In order to encourage the expert to specialize, the discriminator is also explicitly trained against the outputs of the losing experts. Then, a minibatch of canonical MNIST digit is used in order to update the discriminator with 'real' data.

We ran the experiments 10 times with different random seeds for the initializations. Each experiment is run for 2000 iterations.

## 5  RESULTS

The experts correctly specialized on inverting exactly one mechanism each in 7 out of the 10 runs; in the remaining 3 runs the results were only slightly suboptimal: one expert specialized on two tasks, one expert did not specialize on any, and the remaining experts still specialized on one task each, thus still covering all the existing tasks. In Figure 2 we show a randomly selected batch of inputs and corresponding outputs from the model. Each independent mechanism was inverted by a different expert.

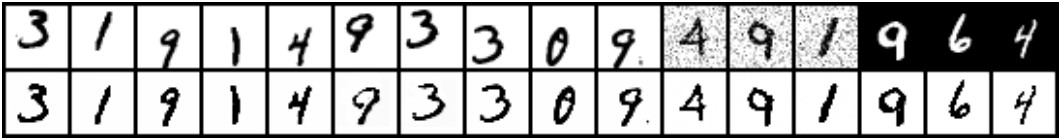

Figure 2: The top row contains 16 random inputs to the networks, and the bottom row the corresponding outputs from the highest scoring experts against the discriminator after 1000 iterations.

First, we discuss the three major aspects of our results followed by additional experiments.

**1. The experts specialize w.r.t. $c$.** We encourage the reader to look at Figure 6 in Appendix A.1, where we plot the scores assigned by the discriminator for each expert on each task in a typical successful run. The figure shows that after an initial chaotic phase of heavy competition, the experts exhibit the desired behavior and obtain a high score on $D$ on one mechanism each. Figure 3 provides further evidence, by visualising that the clusters induced by $c$ are meaningful. We report the proportion of examples from each task assigned to each expert at the beginning and at the end of training. At first, a couple of experts win most examples from all tasks. By the end of the training, each expert wins almost all examples coming from one transformation, and no other.

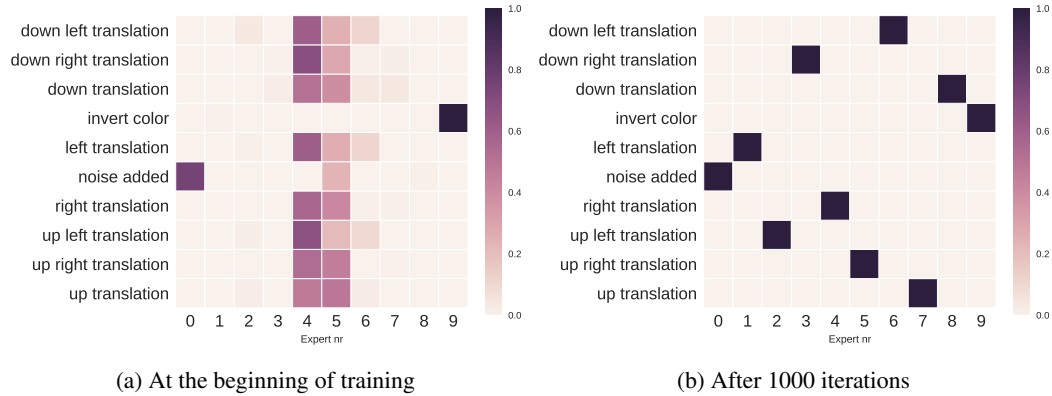

(a) At the beginning of training

(b) After 1000 iterations

Figure 3: The proportion of data won by each expert for each transformation on the digits from the test set.

**2. The transformed outputs improve a classifier.** In order to test if the committee of experts can overall recover a good approximation of the original digits, we test the output of our experts against a pretrained MNIST classifier. For this, we use the test partition of the data. We compute the accuracy for three inputs: *a*) the transformed test digits, *b*) the transformed digits after being processed by the highest scoring experts, *c*) the original test digits. The latter can be seen as an upper bound to the accuracy that can be achieved.

As shown by the two dashed horizontal lines in Figure 4, the transformed test digits achieve a 40% accuracy when tested directly on the classifier, while the untransformed digits would achieve $\approx 99\%$ accuracy. The accuracy for the output digits also starts at 40% — due to the identity initialization — and quickly matches the performance of the original digits as it is trained. Note also that after about 600 iterations — i.e. as the networks have seen overall about one third of the whole dataset, and once only — the accuracy is already almost at the upper bound.

**3. The experts learn mechanisms.** Finally, we test the networks on inputs that were not transformed with the mechanisms that each of them has learned to invert. As shown in Figure 5, each network consistently applies the same transformation also on inputs outside of its training distribution, and therefore the experts not only recovered the correct digits for the domain they have specialized on, but indeed learned the independent *mechanisms*. Since the experts are fully convolutional networks in this experiment, they could be even be ported to other domains with images of different sizes.

**Effect of the approximate identity initialization.**   When running the same experiments without the approximate identity initialization, we found that often several experts fail to specialize. Out of 10 new runs with random initialization, only one experiment had arguably good results, with eight experts specializing on one task each, one expert on two tasks, and the last expert on none. The performance was worse in the remaining runs. We tested whether the problem was that the algorithm takes longer to converge following a random initialization, and ran one additional experiment for 10 000 iterations. The results did not improve.

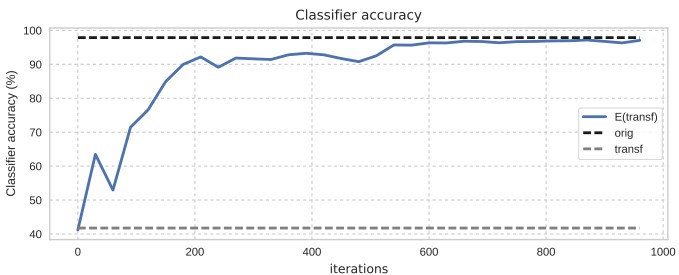

Figure 4: Accuracy on the transformed test digits $\mathcal{D}_Q$ of a pretrained CNN MNIST classifier, on the same digits after going through our model, and on the original digits before transformation $\mathcal{D}_P$ (here $\mathcal{D}_P$ corresponds to the ground truth images in $\mathcal{D}_Q$ for the true applied mechanisms).

Figure 5: Each column shows how each expert transforms the input presented on top. We arrange the tasks such that on the diagonal there is the highest scoring expert for the input given at the top of the column. It is evident that the experts have learned the mechanisms, as they consistently apply them to digits outside of their training domain.

**A simple single-net baseline.** Training a single network instead of a committee of experts makes the problem more difficult to solve. Using identical training settings, we trained a single network once with 32, once with 64, and once with 128 filters per layer, and none of them managed to correctly learn more than one inverse mechanism.[4] Note that a single network with 128 filters per layer has about twice as many parameters overall than the committee of 10 experts with 32 filters per layer each. We also tried random initialization instead of the approximate identity, to reduce the learning rate of the discriminator by a factor of 10, and to increase the receptive field by adding two pooling and two upsampling layers, without any improvement. While we do not exclude that careful hyperparameter tuning may enable a single net to learn multiple mechanisms, it is not entirely straightforward in our experiment.

**Specialization occurs also with higher capacity experts.** While in principle with infinite capacity and data a single expert could solve all tasks simultaneously, in practice limited resources and the proposed training procedure favor specialization in independent modules. Increasing the size of the experts from 32 filters per layer to 64 or 128 filters[5] or enlarging the overall receptive field by using two pooling and two upsampling layers, still results in good specialization of the experts, with no more than two experts specializing on up to two tasks at once.

**Fewer examples from the canonical distribution.** In many applications, we might only have a small sample from the original distribution. Interestingly, we found that all experts still specialize to different tasks and recover good approximations of the inverse mechanisms when we reduce the number of examples from the original distribution from 30 000 down to 64[6]. Even though the output digits are not as clean and sharp, we still achieve 96% accuracy on the pretrained classifier before the discriminator starts to overfit.

---

[4]Specifically, the network performs well on the contrast inversion task, and poorly on all others.

[5]Equivalent to an increase of parameters from $\sim$27K to $\sim$110K or 440K parameters respectively.

[6]Every other aspects of the training remain unchanged.

## 6    CONCLUSIONS

We have developed a method to identify and learn a set of independent causal mechanisms. In the present work, these are inverse mechanisms, but an extension to forward mechanisms appears feasible and worthwhile. We reported promising results in an experiment based on image transformations; future work could study more complex settings and diverse domains.

A natural extension of our work is to consider independent mechanisms that *simultaneously* affect the data (e.g. lighting and position in a portrait), and to allow multiple passes through our committee of experts to identify local mechanisms (akin to Lie derivatives) from more complex datasets — for instance, using recurrent neural network that allow the application of multiple mechanisms by iteration.

Note that for large numbers of experts, the computational cost might become unnecessarily high. This could be mitigated by hybrid approaches incorporating gated mixture of experts — which may exhibit lower computational complexity — or a hierarchical selection of competing experts.

We believe our work constitutes a relevant connection between causal modeling and deep learning. As discussed in the introduction, causality has a lot to offer for crucial machine learning problems such as transfer or compositional modeling. Our systems illustrates some of these properties. Independent modules as sub-components could be trained independently and/or from multiple domains, added subsequently, and transferred to other problems. This may constitute a step towards causally motivated life-long learning.

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

## A   ADDITIONAL RESULTS.

### A.1   PLOT OF COMPETING EXPERTS.

Each expert in Figure 6 is represented with the same color and linestyle across all tasks. Note how the red expert tries to learn two similar tasks until iteration 500 (i.e. left and up-left translation), when the green expert takes over one of the task and they can then both quickly specialize.

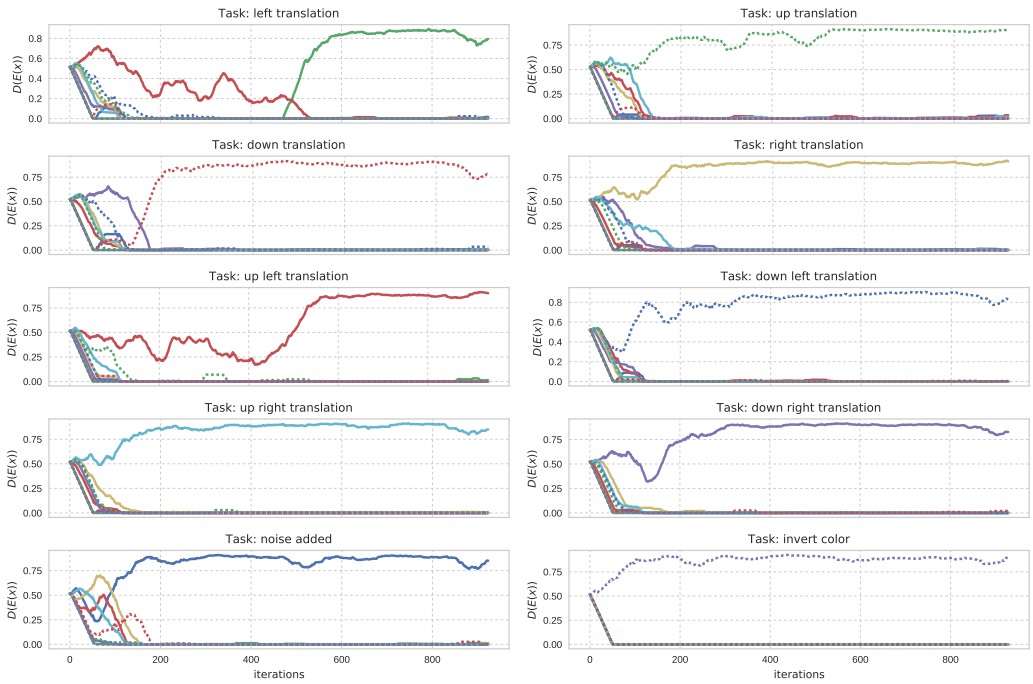

Figure 6: Each line style is associated to the score that an expert obtains on the discriminator when being fed transformed digits using one of the mechanisms. Each expert learns to specialize on a different mechanism. Each curve is smoothed with an average of the last 50 iterations for ease of visualization.

### A.2   TOO MANY OR TOO FEW EXPERTS.

**Too many experts**   When there are too many experts, for most tasks only one wins all the examples, as shown in Figure 7 where the model has 16 experts for 10 tasks. The remaining experts either do not specialize at all — and therefore can be removed from the architecture — or specialize on the same task, and could therefore be combined if after inspection they are considered to perform the same task. Since the accuracy on the transformed data tested on the pretrained classifier reaches again the upperbound of the untransformed data, and since the progress is very similar to that illustrated in Figure 4, we omit this plot.

**Too few experts**   For a committee of 6 experts, the networks do not reconstruct properly most of the digits, which is reflected by an overall low objective function value on the data. Also, the score against the classifier that does not exceed 72%. A few experts are inevitably assigned to multiple tasks, and by looking at Figure 7 it is interesting to see that the clustering result is still meaningful (e.g. expert 5 is assigned to left, down-left, and up-left translation).

## B   DETAILS OF NEURAL NETWORKS

In Table 1 we report the configuration of the neural networks used in these experiments.

## C    TRANSFORMATIONS

In our experiments we use the following transformations

- Translations: the image is shifted by 4 pixels in one of the eight directions up, down, left, right and the four diagonals.

- contrast (or color) inversion: the value of each pixel — originally in the range $[0, 1]$ — is recomputed as $1-$ the original value.

- Noise addition: random Gaussian noise with zero mean and variance 0.25 is added to the original image, which is then clamped again to the $[0, 1]$ interval.

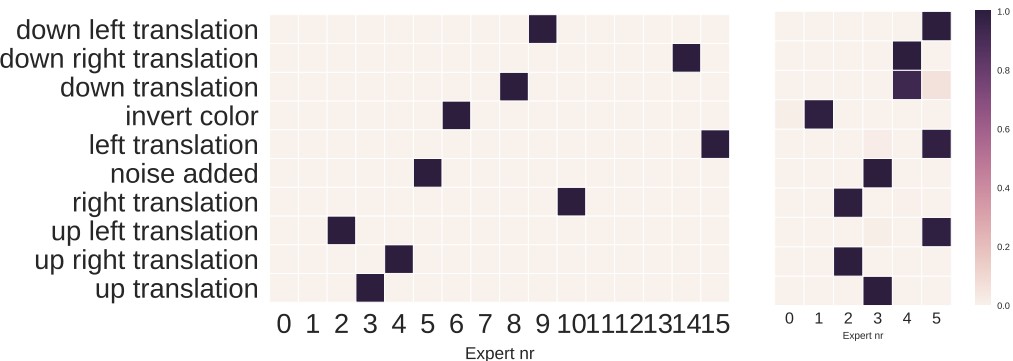

Figure 7: The proportion of data won by each expert for each transformation on the digits from the test set, for the case of 10 mechanisms and more experts (16 on left) or too few (6 on the right).

Table 1: Architectures of the neural networks used in the experiment section. BN stands for Batch normalization, FC for fully connected. All convolutions are preceded by a 1 pixel zero padding.

| Discriminator |
| --- |
| **Layers** |
| $3 \times 3, 16$, ELU |
| $3 \times 3, 16$, ELU |
| $3 \times 3, 16$, ELU |
| $2 \times 2$, avg pooling |
| $3 \times 3, 32$, ELU |
| $3 \times 3, 32$, ELU |
| $2 \times 2$, avg pooling |
| $3 \times 3, 64$, ELU |
| $3 \times 3, 64$, ELU |
| $2 \times 2$, avg pooling |
| $1024$, FC, ELU |
| $1$, FC, sigmoid |

| Expert |
| --- |
| **Layers** |
| $3 \times 3, 32$, BN, ELU |
| $3 \times 3, 32$, BN, ELU |
| $3 \times 3, 32$, BN, ELU |
| $3 \times 3, 32$, BN, ELU |
| $3 \times 3, 1$, sigmoid |

## D    NOTES ON THE FORMALIZATION OF INDEPENDENCE OF MECHANISMS

In this section we briefly discuss the notion of independence of mechanisms as in (Janzing & Schölkopf, 2010), where the independence principle is formalized in terms of algorithmic complexity (also known as Kolmogorov complexity). We summarize the main points needed in the present context. We parametrize each *mechanism* by a bit string $x$. The Kolmogorov complexity $K(x)$ of $x$ is the length of the shortest program generating $x$ on an a priori chosen universal Turing machine.

The **algorithmic mutual information** can be defined as $I(x : y) := K(x) + K(y) - K(x, y)$, and it can be shown to equal

$$I(x : y) = K(y) - K(y|x^*), \tag{4}$$

where for technical reasons we need to work with $x^*$, the shortest description of $x$ (which is in general uncomputable). Here, the conditional Kolmogorov complexity $K(y|x)$ is defined as the length of the shortest program that generates $y$ from $x$. The algorithmic mutual information measures the algorithmic information two objects have in common. We define two mechanisms to be **(algorithmically) independent** whenever the length of the shortest description of the two bit strings together is not shorter than the sum of the shortest individual descriptions (note it cannot be longer), i.e., if their algorithmic mutual information vanishes.[7] In view of (4), this means that

$$K(y) = K(y|x^*). \tag{5}$$

We will say that two mechanisms $x$ and $y$ are independent whenever the complexity of the conditional mechanism $y|x$ is comparable to the complexity of the unconditional one $y$. If, in contrast, the two mechanisms were closely related, then we would expect that we can mimic one of the mechanisms by applying the other one followed by a low complexity conditional mechanism.

This can be implemented by having a complexity measure for, say, neural networks, and comparing the complexities of neural nets that are trained to perform certain tasks. Inspired by regularization theory, we could measure complexity by inverse regularization strength or weight vector norm. An alternative way to regularize neural nets consists of early stopping. If we fix the number of training epochs to a constant, and find that network 1 reaches a lower error than network 2, we conclude that the network 2 would take longer to reach the same low error, and thus network 2 requires higher effective complexity to solve its task than network 1. We have run preliminary experiments with this measure and found that (1) indeed our training procedure did increase independence, and (2) the independence between two different mechanism was larger than the independence between one mechanism and the identity.

---

[7]All statements are valid up to additive constants, linked to the choice of a Turing machine which produces the object (bit string) when given its compression as an input. For details, see (Janzing & Schölkopf, 2010).

