# OpenReview forum: "Learning Independent Causal Mechanisms"
_ICLR.cc/2018/Conference — Reject_

### Official Review · AnonReviewer3 · 2017-11-23
**Review for Learning Independent Causal Mechanisms**

**Rating:** 6
**Confidence:** 4

**Review:**

This paper presents a framework to recover a set of independent mechanisms. In order to do so it uses a set of experts each one made out of a GAN.

My main concern with this work is that I don't see any mechanism in the framework that prevents an expert  (or few of them) to win all examples except its own learning capacities. p7 authors have also noticed that several experts fail to specialize and I bet that is the reason why.
Thus, authors should analyze how well we can have all/most experts specialize in a pool vs expert capacity/architecture.
It would also be great to integrate a direct regularization mechanism in the cost  in order to do so. Like for example a penalty in how many examples a expert has catched.

Moreover, the discrimator D  (which is trained to discriminate between real or fake examples) seems to be directly used to tell if an example is throw from the targeted distribution. It is not the same task. How D will handle an example far from fake or real ones ? Why will D answer negatively (or positively) on this example ?

---

> ### Author Response · Authors · 2018-01-02
> **Answer**
>
> ] My main concern with this work is that I don't see any mechanism in the framework that prevents an expert  (or few of them) to win all examples except its own learning capacities.
>
> This is correct. If the experts have unlimited capacity and the data is unlimited, then a single network can learn the whole thing. Having limited capacity and finite data, on the other hand, favors specialization into independent modules, together with the way we set up the method:
> 1) the experts start from a similar ground (all initialized approximately as identity),
> 2) they compete for data (only the winning expert is trained on a given example),
> 3) the mechanisms are independent
>
> In practice, as a mechanism starts to specialize on one task, it tends to get worse on the other tasks (at least, worse than other modules initialized to the identity). In our experience, experts usually fail to specialize if there are too many experts for the number of mechanisms present in the dataset, or if they are not initialized to the identity, both of which make perfect sense.
>
> ] p7 authors have also noticed that several experts fail to specialize and I bet that is the reason why.
>
> At page 7 the case where several fail is the one where they are initialized randomly (not with identity), which explains why they fail. Winning a few examples in the beginning will make the output of this lucky expert look “better” (more like MNIST digits) than the (random) outputs of the remaining experts. Hence such a “lucky” expert will likely continue to win almost all examples.
>
> ] Thus, authors should analyze how well we can have all/most experts specialize in a pool vs expert capacity/architecture.
>
> We ran new experiments with experts that have more capacity both in terms of number of filters --- 128 instead of 32 --- and in terms of size of the overall receptive field by adding two downsampling (2x2 average pooling) and two upsampling layers (2x2 nearest neighbor).
> For more filters, the results and the training curves are almost identical to the ones obtained with smaller experts, with 9 or 10 experts specializing in every run.
> For a larger receptive field there are still only isolated occurrences of up to two experts trying to specialize on up two tasks each.
>
> ] It would also be great to integrate a direct regularization mechanism in the cost  in order to do so. Like for example a penalty in how many examples a expert has won.
>
> This would be a good way to incorporate prior knowledge on the tasks. In order to do this one would need to know approximately how many mechanisms are at play and what is the prior probability to choose any of them.
> In our setting we assume we do not have such prior knowledge.
>
> ] Moreover, the discrimator D  (which is trained to discriminate between real or fake examples) seems to be directly used to tell if an example is throw from the targeted distribution. It is not the same task. How D will handle an example far from fake or real ones ? Why will D answer negatively (or positively) on this example ?
>
> It is true that in general, even when training standard GANs, the behavior of a discriminator outside of the domain of the real and fake data is undefined, and one should not expect it to be meaningful (not even for the 'perfect' discriminator).
>
> In our case, the discriminator is trained on all outputs from all experts (see page 6: "In order to encourage the expert to specialize, the discriminator is also explicitly trained against the outputs of the losing experts" and in the Algorithm, line 5).
>
> From the standard i.i.d. assumption, we then conclude that the discriminator can be used directly to judge any example coming from any of the experts. If the i.i.d. assumption is invalid, the discriminator will indeed give a meaningless answer, unless training continues on the non-i.i.d. data as well.

---

### Official Review · AnonReviewer2 · 2017-11-26
**limited setting, but exhibits interesting behavior**

**Rating:** 5
**Confidence:** 4

**Review:**

This paper describes a setting in which a system learns collections of inverse-mapping functions that transform altered inputs to their unaltered "canonical" counterparts, while only needing unassociated and separate sets of examples of each at training time.  Each inverse map is an "expert" E akin to a MoE expert, but instead of using a feed-forward gating on the input, an expert is selected (for training or inference) based on the value of a distribution-modeling function c applied to the output of all experts:  The expert with maximum value c(E(x)) is selected.  When c is an adversarially trained discriminator network, the experts learn to model the different transformations that map altered images back to unaltered ones.  This is demonstrated using MNIST with a small set of synthetic translations and noise.

The fact that these different inverse maps arise under these conditions is interesting --- and Figure 5 is quite convincing in showing how each expert generalizes.  However, I think the experimental conditions are very limited:  Only one collection of transformations is studied, and on MNIST digits only.  In particular, I found the fact that only one of ten transformations can be applied at a time (as opposed to a series of multiple transforms) to be restrictive.  This is touched on in the conclusion, but to me it seems fundamental, as any real-world new example will undergo significantly more complex processes with many different variables all applied at once.

Another direction I think would be interesting, is how few examples are needed in the canonical distribution?  For example, in MNIST, could the canonical distribution P be limited to just one example per digit (or just one example per mode / style of digit, e.g. "2" with loop, and without loop)?  The different handwriters of the digits, and sampling and scanning process, may themselves constitute in-the-wild transformations that might be inverted to single (or few) canonical examples --- Is this possible with this mechanism?

Overall, it is nice to see the different inverse maps arise naturally in this setting.  But I find the single setting limiting, and think the investigation could be pushed further into less restricted settings, a couple of which I mention above.



Other comments:

- c is first described to be any distribution model, e.g. the autoencoder described on p.5.  But it seems that using such a fixed, predefined c like the autoencoder may lead to collapse:  What is preventing an expert from learning a single constant mode that has high c value?  The adversarially trained c doesn't suffer from this, because presumably the discriminator will be able to learn the difference between a single constant mode output and the distribution P.  But if this is the case, it seems a critical part of the system, not a simple implementation choice as the text seems to say.

- The single-net baseline is good, but I'd like to get a clearer picture of its results.  p.8 says this didn't manage to "learn more than one inverse mechanism" --- Does that mean it learns to invert a single mechanism (that is always translates up, for example, when presented an image)?  Or that it learned some mix of transforms that didn't seem to generalize as well?  Or does it have some other behavior?  Also, I'm not entirely clear on how it was trained wrt c --- is argmax(c(E(x)) always just the single expert?  Is c also trained adversarially?  And if so, is the approximate identity initialization used?

---

> ### Author Response · Authors · 2018-01-02
> **Answer pt2**
>
> ] - The single-net baseline is good, but I'd like to get a clearer picture of its results.  p.8 says this didn't manage to "learn more than one inverse mechanism" --- Does that mean it learns to invert a single mechanism (that is always translates up, for example, when presented an image)?  Or that it learned some mix of transforms that didn't seem to generalize as well?  Or does it have some other behavior?  Also, I'm not entirely clear on how it was trained wrt c --- is argmax(c(E(x)) always just the single expert?  Is c also trained adversarially?  And if so, is the approximate identity initialization used?
>
> The single net baseline learns a mix of transformations that do not generalize well: while color inversion is often correctly learned, the other mechanisms are mapped to odd looking shapes or digits outlines.
> To answer the specific questions:
> - the argmax in this case is indeed the only net
> - c is still trained adversarially
> - we do use the identity initialization.
>
> Following up on the reviewer's comment we also tried the following extra configurations for the single net baseline:
> - not use the identity initialization
> - reduce the learning rate of the discriminator by a factor of 10
> - add two downsampling (2x2 average pooling) and two upsampling layers (2x2 nearest neighbor)
> None of these improved the performance of the single net baseline.

---

> ### Author Response · Authors · 2018-01-02
> **Answer pt1**
>
> ] The fact that these different inverse maps arise under these conditions is interesting --- and Figure 5 is quite convincing in showing how each expert generalizes.  However, I think the experimental conditions are very limited:  Only one collection of transformations is studied, and on MNIST digits only.  In particular, I found the fact that only one of ten transformations can be applied at a time (as opposed to a series of multiple transforms) to be restrictive.  This is touched on in the conclusion, but to me it seems fundamental, as any real-world new example will undergo significantly more complex processes with many different variables all applied at once.
>
> We believe the way to approach this will be to consider local mechanisms that can be iterated to generate complex transformations. This will require recurrency, and it may be linked to early work on Lie groups in visual perception. It’s an exciting prospect but beyond the scope of this paper. Right now, the paper should not be judged as a complete solution to the problem of image transformations - we think it’s an intriguing direction, but not the final story yet.
>
> ] Another direction I think would be interesting, is how few examples are needed in the canonical distribution?  For example, in MNIST, could the canonical distribution P be limited to just one example per digit (or just one example per mode / style of digit, e.g. "2" with loop, and without loop)?  The different handwriters of the digits, and sampling and scanning process, may themselves constitute in-the-wild transformations that might be inverted to single (or few) canonical examples --- Is this possible with this mechanism?
>
> This is indeed a very interesting question that we have also given some thought to. We believe this will be easier to do once we can combine local mechanisms, hence we have postponed it for the time being.
> In particular, if there are intrinsic factors of variation (such as handwriting style) the problem again becomes one of disentangling simultaneously present transformations, rather than inverting individual mechanisms.
> Concerning sample efficiency, we followed up with a simple experiment: We still obtain very good results with down to 64 images for the canonical distribution (instead of 30k) and still 30k transformed images (roughly 3k independent images per mechanism). The images produced by the experts are not as clean, but the accuracy reached by the pre-trained classifier still increases from 40% to 96% and the experts still specialize on one mechanism each. Of course the discriminator starts to overfit sooner, and the performance decreases if it is trained long enough.
> For even fewer examples (32), we start to observe overfitting of the discriminator before reaching very good performance of the experts.
> We updated the paper with these results about sample sizes.
>
> Overall, it is nice to see the different inverse maps arise naturally in this setting.  But I find the single setting limiting, and think the investigation could be pushed further into less restricted settings, a couple of which I mention above.
>
> ] Other comments:
> ]
> ] - c is first described to be any distribution model, e.g. the autoencoder described on p.5.  But it seems that using such a fixed, predefined c like the autoencoder may lead to collapse:  What is preventing an expert from learning a single constant mode that has high c value?  The adversarially trained c doesn't suffer from this, because presumably the discriminator will be able to learn the difference between a single constant mode output and the distribution P.  But if this is the case, it seems a critical part of the system, not a simple implementation choice as the text seems to say.
>
> We adjust the statement on page 5, since following up on the reviewer's comment we ran experiments with standard autoencoders, confirming the concern of the reviewer. All experts collapsed to output black images (which are usually perfectly reconstructed by an autoencoder).
>
> We expected VAEs not to suffer from the same problem, and indeed they produced promising results, despite still somewhat inferior to an adversarially trained discriminator. Hence while we can use a distribution model that is not adversarially trained, we agree that the standard autoencoder is not a suitable one and changed the text on page 5 accordingly.

---

### Official Review · AnonReviewer1 · 2017-11-27
**Interesting problem setup, more empirical results and discussions on related works would improve the paper**

**Rating:** 5
**Confidence:** 4

**Review:**

Summary:
Given data from a canonical distribution P and data from distributions that are
independent transformations (mechanisms) applied on P, this paper aims to learn
1) those independent transformations; and 2) inverse transformations that map
data from transformed distributions to their corresponding canonical
distribution.

This is achieved by training a mixture of experts, where each expert is assumed to
model a single inverse transformation. Each expert can be seen as the generator
of a conditional GAN. The discriminator is trained to distinguish samples from
the canonical distribution P and those transformed distributions.

Experiments on MNIST data shows that in the end of training, each expert wins
almost all samples from one transformation and no other, which confirms that
each expert model a single inverse transformation.

Comments:
1) Besides samples from distributions that are results of applying independent
mechanisms, samples from the canonical distribution are also required to learn
the model. Are the samples from the canonical distribution always available in
practice? Since the canonical samples are needed for training, this problem
setup seems not to be totally "unsupervised".

2) The authors only run experiments on the MNIST data, where 1) the mechanisms are
simulated and relatively simple, and 2) samples from the canonical distribution
are also available. Did the authors run experiments on other datasets?

3) This work seems to be related to the work on 1) disentangling factors of
variation; and 2) non-linear independent component analysis. Could the authors
add discussions to illustrate the difference between the proposed work and
those topics?

4) This work is motivated from the objective of causal inference, therefore it
might be helpful to add empirical results to show how the proposed method can
be used for causal inference.

---

> ### Author Response · Authors · 2018-01-02
> **Answer**
>
> ] 1) Besides samples from distributions that are results of applying independent
> ] mechanisms, samples from the canonical distribution are also required to learn
> ] the model. Are the samples from the canonical distribution always available in
> ] practice? Since the canonical samples are needed for training, this problem
> ] setup seems not to be totally "unsupervised".
>
>
> We agree that having samples from both the canonical distribution and the transformed distribution helps learn the mechanisms. Note that the examples are not paired to original examples, so the signal is a weak one. Moreover, one could imagine that one could identify a subset of canonical examples from a larger distribution, i.e., that we construct the canonical distribution from a wider distribution containing transformed images.
> Furthermore, when trying to reconstruct a canonical distribution only from the transformed distributions, we found it hard to identify a “good” or unique criterion. For example, if the only transformations are “up” and “up-right”, it’s not clear why there should be a unique location for the “centered” digits.
>
> We encourage the reviewer to look at the similar comment made by reviewer 2, and our answer with new results using a much more limited amount of examples from the canonical distribution (64 instead of 30'000).
>
> ] 2) The authors only run experiments on the MNIST data, where 1) the mechanisms are
> ] simulated and relatively simple, and 2) samples from the canonical distribution
> ] are also available. Did the authors run experiments on other datasets?
>
> We have so far only considered the MNIST problem.
>
> ] 3) This work seems to be related to the work on 1) disentangling factors of
> ] variation; and 2) non-linear independent component analysis. Could the authors
> ] add discussions to illustrate the difference between the proposed work and
> ] those topics?
>
> Indeed there are close relations to the work on disentangling factors of variation and also non-linear ICA. In our work, causal mechanisms play the role of ‘factors of variation’ which we believe is a fruitful view of this problem. We added a discussion about related works on DFV and non-linear ICA to Section 2 of the paper.
> The causal point of view also builds a bridge to the field of domain adaptation.
> Finally, note that our approach currently recovers inverse mechanisms, as independent and modular parts (in our experiments a separate net for each mechanism), while typically in DFV one is interested in obtaining a joint low dimensional representation of the data without explicit separate paths for each factor.
>
>
> ] 4) This work is motivated from the objective of causal inference, therefore it
> ] might be helpful to add empirical results to show how the proposed method can
> ] be used for causal inference.
>
> The approach is not applicable to standard causal structure learning problems in the current form. We do believe, however, that it can play a role for learning multi-task structural causal models whose structure mimics the mechanistic structure of the data generating process, so ultimately it will also be linked to structure learning. Currently, the influence goes the other way round: the causal view inspires our machine learning approach.

---

### Author Response · Authors · 2017-11-26
**Appendix D, Eq. 4, page 12**

On page 12, Appendix D, a minus sign was not rendered on Equation 4 and the line above. The correct equations are:
I( x : y ) := K( x ) + K( y ) - K( x , y )
I( x : y ) = K( y ) - K( y | x )
The typo has been fixed and won't appear in the revision.

---

### Author Response · Authors · 2018-01-02
**General comment to all reviewers**

We thank all three reviewers for their thorough analysis and extremely insightful comments. Following up on their input, we added new content to the paper, ran new experiments, clarified missing points and adjusted misleading statements.

On top of smaller edits, these are the major additions:

* A paragraph in the related work section (Sec. 2) about disentangling factors of variation and non-linear independent component analysis.
* A paragraph discussing new experiments on sample efficiency for the canonical distribution, where we obtained almost identical results using only 64 examples instead of 30'000 (Sec. 5, end of page 8).
* We changed the incorrect claim (in Sec. 3, “Concrete protocol for neural networks”) that standard autoencoders could be used as well for c. However, VAEs do work, confirming that adversarial training is not a necessary component of our method.
* We ran new experiments for different single net baselines --- smaller learning rate for the discriminator, with and without identity initialization, larger receptive field --- confirming that it is not straightforward to learn all tasks with a single net (extending paragraph "A simple single-net baseline" in Sec. 5).
* We ran new experiments for experts with larger capacity, to test how specialization is affected (paragraph "Specialization occurs also with higher capacity experts." in Sec. 5).

We believe the paper has greatly benefited from the reviewers' questions, and we encourage them to read the others' comments as well. We welcome further feedback and are happy to expand our comments if anything remains unclear. Finally, we kindly ask the reviewers to consider re-evaluating their final scores taking into account the new edits and added material.

---

### Decision · Program_Chairs · 2018-01-29
**ICLR 2018 Conference Acceptance Decision**

**Decision:**

Reject

**Comment:**

PROS:
1. All the reviewers thought that the work was interesting and showed promise
2. The paper is relatively well written

CONS:
1. Limited experimental evaluation (just MNIST)

The reviewers were all really on the fence about this but in the end felt that while the idea was a good one and the authors were responsive in their rebuttal, the experimental evaluation needed more work.